# EMab-300 Detects Mouse Epidermal Growth Factor Receptor-Expressing Cancer Cell Lines in Flow Cytometry

**DOI:** 10.3390/antib12030042

**Published:** 2023-06-21

**Authors:** Nohara Goto, Hiroyuki Suzuki, Tomohiro Tanaka, Kenichiro Ishikawa, Tsunenori Ouchida, Mika K. Kaneko, Yukinari Kato

**Affiliations:** 1Department of Molecular Pharmacology, Tohoku University Graduate School of Medicine, 2-1 Seiryo-machi, Aoba-ku, Sendai 980-8575, Miyagi, Japan; s1930550@s.tsukuba.ac.jp (N.G.); tomohiro.tanaka.b5@tohoku.ac.jp (T.T.); ken.ishikawa.r3@dc.tohoku.ac.jp (K.I.); k.mika@med.tohoku.ac.jp (M.K.K.); 2Department of Antibody Drug Development, Tohoku University Graduate School of Medicine, 2-1 Seiryo-machi, Aoba-ku, Sendai 980-8575, Miyagi, Japan; tsunenori.ouchida.d5@tohoku.ac.jp

**Keywords:** mouse EGFR, monoclonal antibody, Cell-Based Immunization and Screening (CBIS)

## Abstract

Epidermal Growth Factor Receptor (EGFR) overexpression or its mutation mediates the sustaining proliferative signaling, which is an important hallmark of cancer. Human EGFR-targeting monoclonal antibody (mAb) therapy such as cetuximab has been approved for clinical use in patients with colorectal cancers and head and neck squamous cell carcinomas. A reliable preclinical mouse model is essential to further develop the mAb therapy against EGFR. Therefore, sensitive mAbs against mouse EGFR (mEGFR) should be established. In this study, we developed a specific and sensitive mAb for mEGFR using the Cell-Based Immunization and Screening (CBIS) method. The established anti-mEGFR mAb, EMab-300 (rat IgG_1_, kappa), reacted with mEGFR-overexpressed Chinese hamster ovary-K1 (CHO/mEGFR) and endogenously mEGFR-expressed cell lines, including NMuMG (a mouse mammary gland epithelial cell) and Lewis lung carcinoma cells, using flow cytometry. The kinetic analysis using flow cytometry indicated that the *K*_D_ of EMab-300 for CHO/mEGFR and NMuMG was 4.3 × 10^−8^ M and 1.9 × 10^−8^ M, respectively. These results indicated that EMab-300 applies to the detection of mEGFR using flow cytometry and may be useful to obtain the proof of concept in preclinical studies.

## 1. Introduction

The Epidermal Growth Factor Receptor (EGFR) belongs to the ERBB family of receptor tyrosine kinases. EGFR is a type I transmembrane glycoprotein, which is composed of an extracellular ligand-binding domain and an intracellular tyrosine kinase domain [1]. Upon ligand binding to the extracellular domain of EGFR, the downstream signaling pathways, such as the mitogen-activated protein kinases, the phosphoinositide 3-kinase/Akt, the Janus kinase/signal transducer and activator of transcription, and the phospholipase C-γ/protein kinase C pathways are activated [2]. These pathways lead to the transcriptional activation of target genes, which are important for cell proliferation, survival, migration, invasion, and cancer-initiating properties [2]. In many carcinomas, EGFR overexpression and its mutation are involved in sustaining proliferative signaling, which is an important hallmark of cancer [3,4]. Therefore, EGFR has been considered as an important target for cancer therapy.

The EGFR-targeted therapies have been used in the clinic, including monoclonal antibodies (mAbs) [5] and the tyrosine kinase inhibitors [6]. Cetuximab is a mouse/human chimeric mAb that binds to the extracellular domain (domain III) of EGFR, which is important for neutralization activity [7]. Cetuximab has been approved by the Food and Drug Administration for colorectal cancer [8] and head and neck squamous cell carcinoma (HNSCC) [9]. Although cetuximab has been used to treat patients with metastatic colorectal cancers, the use is limited to tumors harboring wild-type RAS [10]. The cetuximab treatment did not exhibit a benefit in patients with colorectal cancer harboring RAS mutations [10]. Moreover, the majority of HNSCC express EGFR; however, the benefit of therapy is limited to only 15–20% of HNSCC patients [11].

Preclinical mouse models have been developed for the establishment of cancer therapy. The earliest models were built through the transplantation of murine tumors into immunocompetent host mice [12]. Furthermore, tumors harvested from genetically modified mice can be transplanted and expanded into fully immunocompetent syngeneic hosts [13]. These syngeneic models were used in preclinical studies to evaluate not only small-molecule chemotherapeutic drugs but also immunotherapies including immune checkpoint inhibitors [13]. Although several studies have shown that cetuximab can stimulate the innate or adaptive immune responses [14,15], the mechanisms have not been fully understood, due to the lack of a preclinical model using anti-mouse EGFR (mEGFR) mAb. The model is also thought to be important for the evaluation of novel antibody–drug conjugates and prediction of the side-effects.

The Cell-Based Immunization and Screening (CBIS) method includes immunization with antigen-overexpressing cells and high-throughput hybridoma screening using flow cytometry. Using the CBIS method, we have developed mAbs against human antigens, including human epidermal growth factor receptor 2 (HER2) [16], human epidermal growth factor receptor 3 (HER3) [17], epithelial cell adhesion molecule (EpCAM) [18,19], trophoblast cell surface antigen 2 (TROP2) [20,21], programmed cell death ligand 1 (PD-L1) [22], podoplanin (PDPN) [23,24,25,26,27,28,29,30,31,32,33,34], the cluster of differentiation 19 (CD19) [35], CD20 [36,37], CD44 [38,39,40,41], CD133 [42], killer cell lectin-like receptor G1 (KLRG1) [43], C-C motif chemokine receptor 9 (CCR9) [44], and T cell immunoreceptor with Ig and ITIM domains (TIGIT) [45] by the immunization of antigen-overexpressing cells in mice. We also successfully developed mAbs against mouse antigens, including mouse CCR3 [46] and mouse CCR8 [47], by their immunization in rats. In this study, we developed novel anti-mEGFR mAbs using the CBIS method and evaluated its applications including flow cytometry.

## 2. Materials and Methods

### 2.1. Preparation of Plasmids

The expression plasmid of mEGFR (pCMV6-neo-mEGFR-Myc-DDK) is commercially available from OriGene Technologies, Inc. (Rockville, MD, USA). The cDNA encoding mEGFR (NM_207655) was subcloned into pCAG-zeo_ssnPA and pCAG-zeo_MAP vectors, which were purchased from FUJIFILM Wako Pure Chemical Corporation (Osaka, Japan), with the N-terminal PA tag [48,49,50] and MAP tag [51,52], respectively. The amino acid sequences of the tag system were as follows: PA tag, 12 amino acids (GVAMPGAEDDVV) and MAP tag, 12 amino acids (GDGMVPPGIEDK). The PA tag can be recognized by NZ-1 (an anti-human PDPN mAb) [48,49,50,53,54,55,56,57,58,59,60,61,62,63,64,65].

### 2.2. Antibodies

Alexa-Fluor-488-conjugated anti-rat IgG was purchased from Cell Signaling Technology, Inc. (Danvers, MA, USA).

### 2.3. Cell Lines

P3X63Ag8U.1 (P3U1), Chinese hamster ovary (CHO)-K1, LN229, and NMuMG (a mouse mammary gland epithelial cell) were obtained from the American Type Culture Collection (ATCC; Manassas, VA, USA). Lewis lung carcinoma was obtained from the Japanese Collection of Research Bioresources (JCRB; Osaka, Japan).

The pCAG-zeo_ssnPA-mEGFR and pCAG-zeo_MAP-mEGFR plasmids were transfected into LN229 and CHO-K1 cells, respectively. The stable transfectants were generated as described previously [40,41].

CHO-K1, mEGFR-overexpressed CHO-K1 (CHO/mEGFR), Lewis lung carcinoma, and P3U1 were cultured in an RPMI-1640 medium (Nacalai Tesque Inc., Kyoto, Japan), with 10% fetal bovine serum (FBS; Thermo Fisher Scientific Inc., Waltham, MA, USA). A cocktail of 100 units/mL of penicillin, 100 μg/mL of streptomycin, and 0.25 μg/mL of amphotericin B (Nacalai Tesque Inc.) was added to the medium. LN229, mEGFR-overexpressed LN229 (LN229/mEGFR), and NMuMG were cultured in DMEM (Nacalai Tesque Inc.), supplemented as indicated above. For NMuMG cells, 10 μg/mL of insulin (Sigma-Aldrich Corp., St. Louis, MO, USA) was further added. All cells were cultured using a humidified incubator at 37 °C, in an atmosphere of 5% CO_2_ and 95% air.

### 2.4. Development of Hybridomas

A five-week-old Sprague–Dawley rat was purchased from CLEA Japan (Tokyo, Japan). The animal was housed under specific pathogen-free conditions. All animal experiments were approved by the Animal Care and Use Committee of Tohoku University (Permit number: 2022MdA-001).

To develop mAbs against mEGFR, we intraperitoneally immunized one rat with LN229/mEGFR (1 × 10^9^ cells) plus Imject Alum (Thermo Fisher Scientific Inc.). Inducing a strong immune response with immunogens can be a slow and inefficient process. Adding an adjuvant, such as Imject Alum, to the antigen stimulates an improved immune response compared to the antigen alone. Adjuvants increase the immune response by localizing antigens for an extended time and attracting the appropriate cells to interact with the immunogen and each other. Adjuvants are mixed and injected along with antigens to prevent catabolism. Alum is frequently used as an alternative to Freund’s adjuvants because alum is less hazardous. Although Freund’s complete and incomplete adjuvants produce a stronger, longer-lasting immunogenic response compared to other adjuvants, they are hazardous to the researcher and can produce lesions at the injection site.

After three additional injections every week (1 × 10^9^ cells/rat), a final booster injection (1 × 10^9^ cells/rat) was performed two days before harvesting spleen cells. The hybridomas were produced, as described previously [46]. The hybridoma supernatants were screened using flow cytometry using CHO/mEGFR, CHO-K1, and NMuMG.

### 2.5. Purification of EMab-300

The cultured supernatant of EMab-300 hybridomas was applied to 1 mL of Ab-Capcher (ProteNova, Kagawa, Japan). Ab-capcher is a gel carrier in which the alkali-resistant antibody-binding protein Protein A-R28, developed by Protenova’s patented technology, was immobilized at multiple points at high density. After washing with phosphate-buffered saline (PBS), the antibodies were eluted with an IgG elution buffer (Thermo Fisher Scientific Inc.). Finally, the eluates were concentrated, and the elution buffer was replaced with PBS using Amicon Ultra (Merck KGaA, Darmstadt, Germany).

### 2.6. Flow Cytometric Analysis

CHO/mEGFR, CHO-K1, NMuMG, and Lewis lung carcinoma were harvested after a brief exposure to 1 mM ethylenediaminetetraacetic acid (EDTA, Nacalai Tesque Inc.). The cells were treated with EMab-300 or blocking buffer (control) (0.1% BSA in PBS) for 30 min at 4 °C, followed by treatment with Alexa-Fluor-488-conjugated anti-rat IgG. The data were analyzed using the SA3800 Cell Analyzer and SA3800 software ver. 2.05 (Sony Corp., Tokyo, Japan).

### 2.7. Determination of K_D_ using Flow Cytometry

We prepared 656 to 0.08 nM (diluted by 1/2) of EMab-300. The serially diluted EMab-300 was suspended with CHO/mEGFR and NMuMG cells for 30 min at 4 °C. The cells were treated with 50 μL of Alexa Fluor 488-conjugated anti-rat IgG (1:200). The fluorescence data were collected using the SA3800 Cell Analyzer. The *K*_D_ was subsequently calculated using GraphPad PRISM 8 (GraphPad Software Inc., La Jolla, CA, USA).

## 3. Results

### 3.1. Development of Anti-mEGFR mAbs Using the CBIS Method

To develop anti-mEGFR mAbs, one rat was immunized with LN229/mEGFR cells (Figure 1A). The spleen was then excised from the rat, and splenocytes were fused with myeloma P3U1 cells (Figure 1B). The developed hybridomas were subsequently seeded into 96-well plates and cultivated for six days. The positive wells were screened by the selection of mEGFR-expressing cell-reactive and CHO-K1-non-reactive supernatants, using flow cytometry (Figure 1C). After the limiting dilution and several additional screenings, an anti-mEGFR mAbs, EMab-300 (rat IgG_1_, kappa), was finally established (Figure 1D).

### 3.2. Flow Cytometric Analysis Using EMab-300

We conducted flow cytometry using EMab-300 against CHO/mEGFR, NMuMG, and Lewis lung carcinoma cell lines. EMab-300 recognized CHO/mEGFR cells dose-dependently at 10, 1, 0.1, and 0.01 μg/mL (Figure 2A). Parental CHO-K1 cells were not recognized even at 10 μg/mL of EMab-300 (Figure 2B).

Next, we investigated the reactivity of EMab-300 against endogenously mEGFR-expressing cell lines, NMuMG and Lewis lung carcinoma. EMab-300 reacted with NMuMG and Lewis lung carcinoma in a dose-dependent manner (Figure 2C,D). These results suggested that EMab-300 specifically recognizes mEGFR, and it is also useful for detecting endogenous mEGFR using flow cytometry.

### 3.3. Kinetic Analysis of EMab-300 Using Flow Cytometry

To determine the *K*_D_ of EMab-300 with mEGFR-expressing cells, we conducted kinetic analyses by flow cytometry using CHO/mEGFR and NMuMG cells. The *K*_D_ values of EMab-300 for CHO/mEGFR and NMuMG were determined as 4.3 × 10^−8^ M and 1.9 × 10^−8^ M, respectively (Figure 3A,B). These results indicate that EMab-300 possesses a moderate affinity for both CHO/mEGFR and NMuMG cells.

## 4. Discussion

Revolutionary therapeutic strategies and modalities for cancer have been developed. However, only about five percent of new cancer therapies are approved, and most fail due to the lack of efficacy [13]. These failures cost significant amounts of money and reduce the patient’s quality of life. The failures also indicate that current preclinical methods are not sufficient to predict successful outcomes. Although EGFR is one of the important targets for cancer therapy, anti-mEGFR mAbs for preclinical study have not been developed. In this study, we developed a novel anti-mEGFR mAb (EMab-300) using the CBIS method and showed the application to flow cytometry (Figure 2 and Figure 3). EMab-300 could contribute to the preclinical study to evaluate the antitumor effects and predict the side-effects of the EGFR-targeting therapy. Furthermore, EMab-300 could be useful to obtain the proof of concept in EGFR-targeting antibody–drug conjugates or combination therapy with immune checkpoint inhibitors in an immunocompetent syngeneic mouse model.

EMab-300 could react with NMuMG and Lewis lung carcinoma cells (Figure 2). NMuMG has been used in the study of the epithelial-to-mesenchymal transition (EMT), which is induced by various cytokines, such as TGF-β [66] and transcriptional factors [67]. The activation of the EMT program confers tumor cells the ability of migration, invasion, extravasation, and stemness [68]. Once the EMT-induced tumor cells reach distant organs, these mesenchymal properties revert to epithelial properties via the mesenchymal–epithelial transition (MET) in order to form a secondary tumor in distant organs [68]. The EMT-induced NMuMG cells can make spheres in the presence of EGF in vitro, and exhibit the tumorigenic potential in vivo [69]. Therefore, we can evaluate the neutralization activity of EMab-300 in the sphere formation assay in a future study.

Lewis lung carcinoma has been widely used as a syngenetic model [70]. The model was a successful preclinical model for the evaluation of a chemotherapeutic agent, navelbine, prior to its implementation in clinical trials [70]. Recently, syngenetic models have also been important for the evaluation of combination therapy with immune checkpoint inhibitors [71,72]. Therefore, the conversion of EMab-300 (rat IgG_1_, kappa) to mouse IgG is essential for evaluating antitumor effects and developing antibody–drug conjugates or chimeric antigen receptor T cell therapies. We previously produced recombinant antibodies, which were converted into the mouse IgG_2a_ subclass from mouse IgG_1_. Furthermore, we produced defucosylated IgG_2a_ mAbs using fucosyltransferase 8-deficient CHO-K1 cells to enhance the antibody-dependent cellular cytotoxicity activity [73,74,75,76,77,78,79,80]. The defucosylated mAbs showed an antitumor effect in xenograft models. Therefore, a class-switched and defucosylated type of EMab-300 could contribute to the treatment of syngenetic mouse tumors and spontaneous mice tumors in the future.

The identification of the epitope is also important to assess the property of mAbs. We established anti-human EGFR mAbs using the CBIS method. Most of the mAbs recognized the conformational epitopes. Therefore, we developed the RIEDL insertion for epitope mapping (REMAP) method [81,82,83,84] to identify the conformational epitope. We determined the conformational epitopes of anti-human EGFR mAbs (EMab-51 and EMab-134) [82,84] and anti-CD44 mAbs (C_44_Mab-5 and C_44_Mab-46) [81,83] using the REMAP method. Therefore, further investigations are required to determine the epitope of EMab-300.

## Figures and Tables

**Figure 1 antibodies-12-00042-f001:**
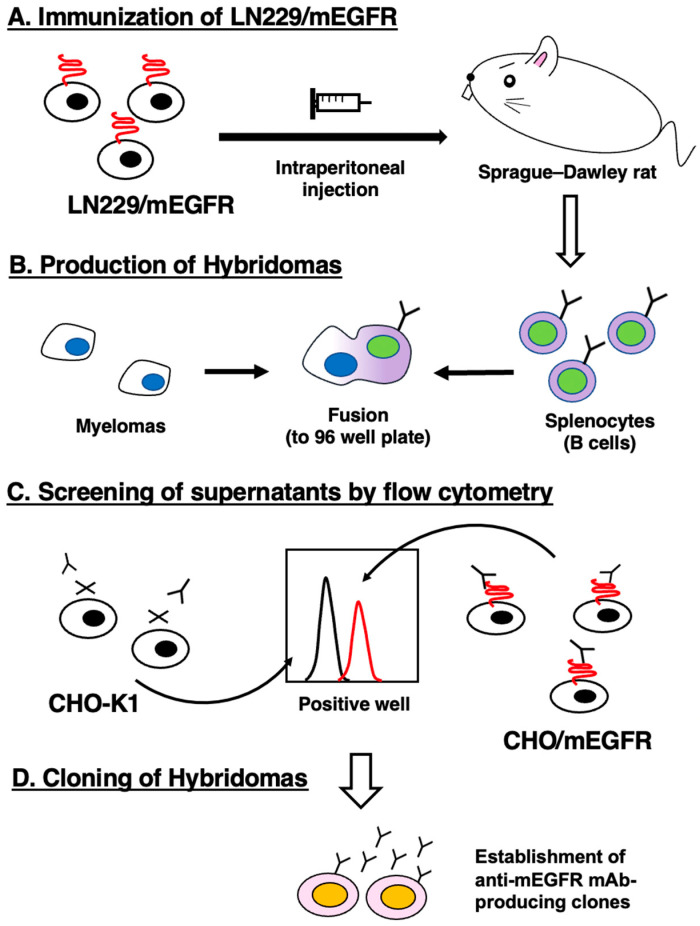
The production of anti-mEGFR mAb, EMab-300. (**A**) LN229/mEGFR cells were immunized into a Sprague–Dawley rat. (**B**) The spleen cells were fused with P3U1 cells. (**C**) To select anti-mEGFR mAb-producing hybridomas, the supernatants were screened by flow cytometry using CHO-K1 and CHO/mEGFR cells. (**D**) After limiting dilution, an anti-mEGFR mAb, EMab-300 was finally established.

**Figure 2 antibodies-12-00042-f002:**
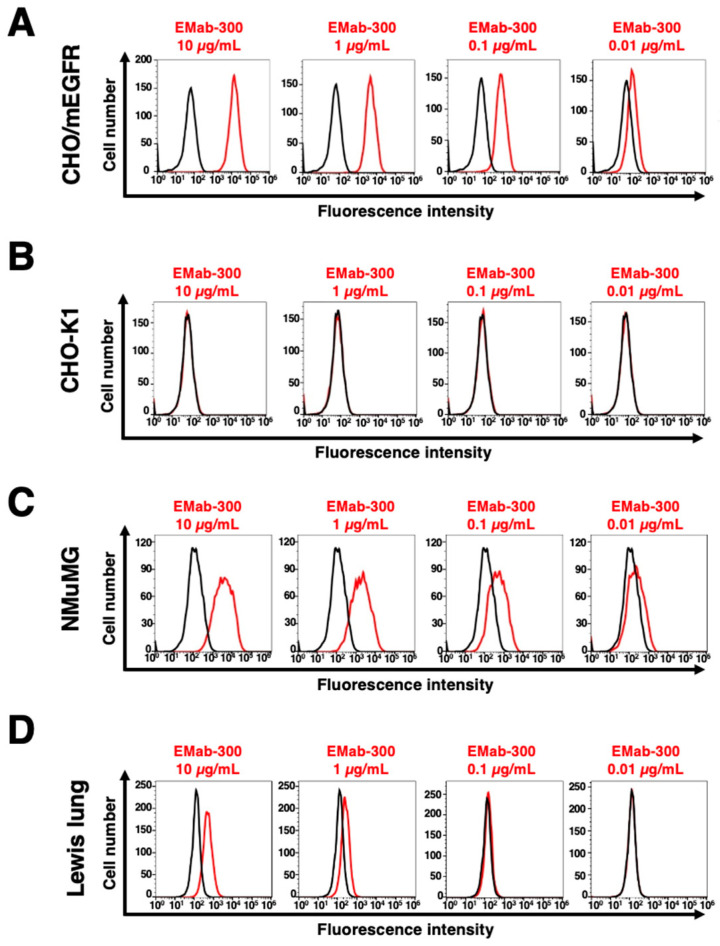
Flow cytometry of mEGFR-expressing cells using EMab-300. CHO/mEGFR (**A**), CHO-K1 (**B**), NMuMG (**C**), and Lewis lung carcinoma (**D**) cells were treated with 0.01–10 µg/mL of EMab-300, followed by treatment with anti-rat IgG conjugated with Alexa Fluor 488. The black line represents the negative control.

**Figure 3 antibodies-12-00042-f003:**
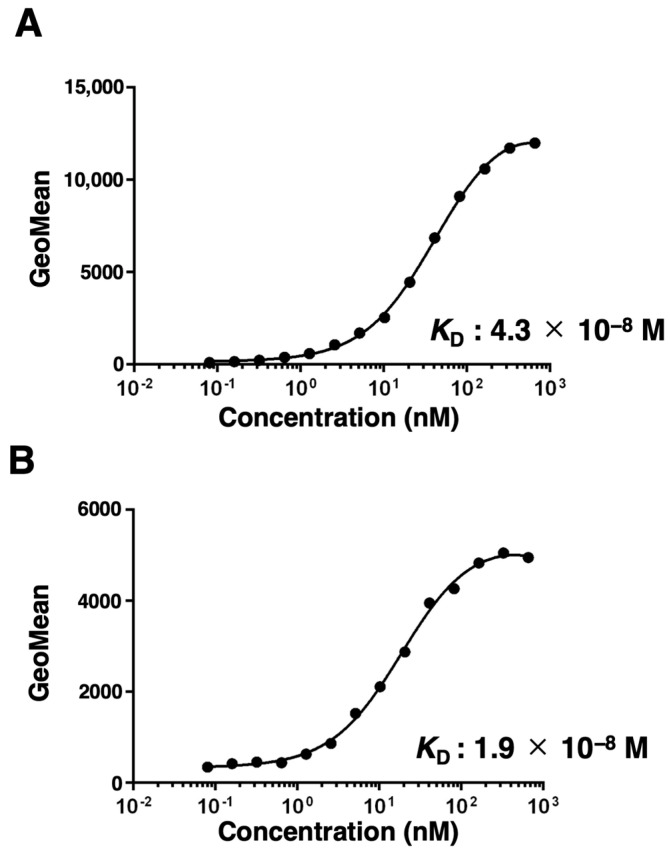
The binding affinity of EMab-300. CHO/mEGFR (**A**) and NMuMG (**B**) cells were suspended in serially diluted EMab-300 as described in “Materials and methods”. The cells were treated with anti-rat IgG conjugated with Alexa Fluor 488. The fluorescence data were subsequently collected using the SA3800 Cell Analyzer, followed by the calculation of the *K*_D_ using GraphPad PRISM 8.

## Data Availability

All related data and methods are presented in this paper. Additional inquiries should be addressed to the corresponding authors.

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
