# Peer review of "EMab-300 Detects Mouse Epidermal Growth Factor Receptor-Expressing Cancer Cell Lines in Flow Cytometry"

_2073-4468, 2023, doi:10.3390/antib12030042_

Round 1
Reviewer 1 Report
In this communication, Goto et al. describe the discovery and validation of a rat antibody specific for mouse EGFR. A rat was immunized with transfected LN229 cells, and then splenocytes were used to generate hybridoma cells, including some specific for mouse EGFR. The resulting antibody EMab-300 was shown to bind to transfected cells and cells naturally expressing mouse EGFR. While the methods and results are clear, the antibody characterization is basic, and the novelty of this work is uncertain.
Major points:
- The abstract states that “a mAb against mouse EGFR for flow cytometry has not been established.” However, a search on Biocompare.com reveals many such antibodies. This language should be updated to reflect existing antibodies.
- The introduction states that cetuximab lacks thorough preclinical data, with the implication that cetuximab does not cross-react with mouse EGFR. However, the details are not described. The cross-reactivity of cetuximab or other clinical antibodies (affinity for mouse EGFR compared to human EGFR) should be defined within the results, or else cited from the literature.
- If EMab-300 is to be a surrogate antibody for cetuximab or other antibodies binding human EGFR, it should bind with similar affinity. But experiments or citations are not provided. What is the relative affinity of cetuximab for human EGFR and EMab-300 for mouse EGFR?
- The epitope of an antibody is an important determinant of its biological effects. If EMab-300 is to be a reliable surrogate antibody for cetuximab, it should bind the same epitope, in addition to having similar affinity. Would it be possible to map the epitope of EMab-300, or at least do an experiment to show whether it competes with cetuximab for binding?
- The results describe flow cytometry experiments investigating the affinity of EMab-300 for EGFR-bearing cells. The term KD is used to describe the cell binding results; but KD is a fundamental biophysical property of antibodies that is independent of cell line or system. It seems more accurate to call the flow cytometry parameter EC50, rather than KD.
Minor points:
- On line 14, it may be more general to state that “its mutation mediates sustaining proliferative signaling in many cancers.”
- On line 25, “and is expected in the use of preclinical study” could be replaced with “may be useful for preclinical studies.”
- On line 64, it should be “immunization with antigen-expressing cells,” not “immunization of antigen-expressing cells.”
- Line 186 could be reworded to, “Revolutionary therapeutic strategies and modalities for cancer have been developed.”
Moderate editing for English language may be required.
Reviewer 2 Report
In the article “EMab-300 Detects Mouse Epidermal Growth Factor Receptor-2 expressing cancer cell lines in flow cytometry”, the authors reported the generation and the selection of new antibody (rat IgG1, kappa) against mouse EGFR using the CBIS method.
Their purpose is to develop a anti-mEGFR antibody to employ for setting mouse preclinical models able to evaluate the antitumor effects and side effects of biomolecules targeting EGFR.
The selected antibody works in flow cytometry using mEGFR-overexpressed Chinese ham-20 ster ovary-K1 (CHO/mEGFR) and endogenously mEGFR-expressed cell lines, including NMuMG 21 (a mouse mammary gland epithelial cell) and Lewis lung carcinoma cells.
The article is well-written and the data are well-presented.
In the discussion, the authors reported some relevant observations such as the need to identify the epitope and assess antibody functionality that should be addressed in future.
I suggest also to plan biochemical assay such as BLI or SPR label free assay to better characterize the antibody in term of affinity constant and kinetics parameters and epitope mapping
the quality of the language is good
Author Response
In the article “EMab-300 Detects Mouse Epidermal Growth Factor Receptor-2 expressing cancer cell lines in flow cytometry”, the authors reported the generation and the selection of new antibody (rat IgG1, kappa) against mouse EGFR using the CBIS method.
Their purpose is to develop a anti-mEGFR antibody to employ for setting mouse preclinical models able to evaluate the antitumor effects and side effects of biomolecules targeting EGFR.
The selected antibody works in flow cytometry using mEGFR-overexpressed Chinese ham-20 ster ovary-K1 (CHO/mEGFR) and endogenously mEGFR-expressed cell lines, including NMuMG 21 (a mouse mammary gland epithelial cell) and Lewis lung carcinoma cells.
The article is well-written and the data are well-presented.
In the discussion, the authors reported some relevant observations such as the need to identify the epitope and assess antibody functionality that should be addressed in future.
I suggest also to plan biochemical assay such as BLI or SPR label free assay to better characterize the antibody in term of affinity constant and kinetics parameters and epitope mapping
Thank you very much for your suggestions.
We have done the identification of epitope for our established anti-human EGFR mAbs including EMab-134. The mAbs were also established by the CBIS method and recognized the conformational epitope. In other words, the mAbs cannot recognize the synthetic peptides derived from human EGFR. Therefore, we identified an epitope of EMab-134 by a novel epitope mapping method, RIEDL insertion for epitope mapping (REMAP) method (MONOCLONAL ANTIBODIES IN IMMUNODIAGNOSIS AND IMMUNOTHERAPY 40, p191-195, 2021).
In the case of EMab-300, we will employ the above strategy to identify the epitope. If we can determine the epitope, we could determine the kinetics parameters using BLI or SPR.
Round 2
Reviewer 1 Report
Thank you for the detailed response to the previous reports, and for the updates to the manuscript.
Upon review, it still seems like there may be other commercial antibodies specific for the extracellular domain of mouse EGFR. Some of these are listed below. While it is possible that some or all of these may not work perfectly in flow cytometry experiments, I still think the wording "a mAb against mouse EGFR (mEGFR) for flow cytometry has not been established" is inaccurate. I would strongly suggest changing this language in the abstract (and also in the conclusions) to recognize the existing antibodies targeting mouse EGFR.
- https://www.novusbio.com/products/egfr-antibody-icr10_nb600-724pe?utm_source=biocompare&utm_medium=referral&utm_campaign=product_NB600-724PE&utm_term=primaryantibodies&utm_content=editorial#datasheet
- https://www.cellsignal.com/product/productDetail.jsp?productId=54359&utm_medium=b2b&utm_campaign=general
- https://www.creative-diagnostics.com/Anti-EGFR-MAb-FITC-Conjugated-160342-491.htm
Moderate editing of English language may be required.
Author Response
Thank you for the detailed response to the previous reports, and for the updates to the manuscript.
Upon review, it still seems like there may be other commercial antibodies specific for the extracellular domain of mouse EGFR. Some of these are listed below. While it is possible that some or all of these may not work perfectly in flow cytometry experiments, I still think the wording "a mAb against mouse EGFR (mEGFR) for flow cytometry has not been established" is inaccurate. I would strongly suggest changing this language in the abstract (and also in the conclusions) to recognize the existing antibodies targeting mouse EGFR.
- https://www.novusbio.com/products/egfr-antibody-icr10_nb600-724pe?utm_source=biocompare&utm_medium=referral&utm_campaign=product_NB600-724PE&utm_term=primaryantibodies&utm_content=editorial#datasheet
- https://www.cellsignal.com/product/productDetail.jsp?productId=54359&utm_medium=b2b&utm_campaign=general
- https://www.creative-diagnostics.com/Anti-EGFR-MAb-FITC-Conjugated-160342-491.htm
Thank you for the suggestion.
According to the comments, we changed the statement in introduction.
Line 17
However, a mAb against mouse EGFR (mEGFR) for flow cytometry has not been established.
-->
Therefore, sensitive mAbs against mouse EGFR (mEGFR) should be established.